# *Chlamydia* Infection Remodels Host Cell Mitochondria to Alter Energy Metabolism and Subvert Apoptosis

**DOI:** 10.3390/microorganisms11061382

**Published:** 2023-05-24

**Authors:** Heng Choon Cheong, Sofiah Sulaiman, Chung Yeng Looi, Li-Yen Chang, Won Fen Wong

**Affiliations:** 1Department of Medical Microbiology, Faculty of Medicine, Universiti Malaya, Kuala Lumpur 50603, Malaysia; cheonghengchoon@gmail.com (H.C.C.); changliyen@um.edu.my (L.-Y.C.); 2Department of Obstetrics and Gynaecology, Faculty of Medicine, Universiti Malaya, Kuala Lumpur 50603, Malaysia; sofiah@um.edu.my; 3School of Biosciences, Faculty of Health and Medical Sciences, Taylor’s University, Subang Jaya 47500, Selangor, Malaysia; chungyeng.looi@taylors.edu.my

**Keywords:** *Chlamydia trachomatis*, mitochondria, sexually transmitted infection, host–pathogen interaction, metabolism, apoptosis

## Abstract

*Chlamydia* infection represents an important cause for concern for public health worldwide. Chlamydial infection of the genital tract in females is mostly asymptomatic at the early stage, often manifesting as mucopurulent cervicitis, urethritis, and salpingitis at the later stage; it has been associated with female infertility, spontaneous abortion, ectopic pregnancy, and cervical cancer. As an obligate intracellular bacterium, *Chlamydia* depends heavily on host cells for nutrient acquisition, energy production, and cell propagation. The current review discusses various strategies utilized by *Chlamydia* in manipulating the cell metabolism to benefit bacterial propagation and survival through close interaction with the host cell mitochondrial and apoptotic pathway molecules.

## 1. Introduction

*Chlamydia* is an ancient infectious disease with the earliest depiction of its infection stretching as far back as 1500 BCE in the Egyptian medical compendium Ebers papyrus [1]. The genus and species designations of the organism are derivatives of the words *chlamys* and *trachys* in the Greek language, which respectively translate into “cloak” and “rough”. Ludwig Halberstädter and Stanislaus von Prowazek coined the genus name in 1907, which stemmed from their observations that specimens of conjunctiva collected from infected orangutan contained inclusions that appeared clustered around the nucleus [2]. The species designation was coined to reflect the clinical appearance of trachoma, which is characterized by roughened eyelids in patients [3,4,5]. Ensuing discoveries of the same intracytoplasmic inclusions in 1911 from children suffering from nongonococcal conjunctivitis, as well as scrapings obtained from their parents with urethritis and cervicitis, would later establish *C. trachomatis* as the etiologic agent of trachoma and genital infections [6]. The bacteria were at one time considered to be a virus owing to their ability to pass through filtration system and their inability to propagate on artificial growth media [7]. The debate about the pathogen’s identity was eventually ended following assignment of the infectious entity to the genus *Chlamydia* [7,8].

*C. trachomatis* is an obligate intracellular Gram-negative pathogen in the family *Chlamydiaceae* that predominantly infects the mucosal epithelia of the reproductive tract, leading to sexually transmitted disease in humans [9]. Alongside trichomoniasis, gonorrhea, and syphilis, genital *C. trachomatis* infection is one of the four most common curable sexually transmitted infections (STIs) in the world, with a worldwide incidence of more than 131 million infections every year [10]. Genital chlamydial infection is prevalent in the urban population, especially among young female adults and prostitutes [11,12]. The prevalence is also relatively higher among females with gynecological issues including infertility, endometriosis, and abnormal menstruation [13]. An increasing trend of *C. trachomatis* STIs has been reported worldwide in recent years, especially among young adolescents. This has raised severe public health concern as an efficient prophylactic vaccine is still not available at present. Chlamydial infections of the genital tract in females are characterized by a vast spectrum of genital tract pathologies that include mucopurulent cervicitis, urethritis, and salpingitis [9,14,15]. The silent nature of the predominant cases of infection has hampered early detection [16]. Lingering infection and its subsequent spread to the upper genital tract are linked to the development of pelvic inflammatory disease (PID), which is a known risk factor for infertility, ectopic pregnancy, and cervical cancer [17,18].

There are currently a total of 19 different *C. trachomatis* serovars known, distinguished by serotype characterization of the chlamydial major outer membrane protein (OmpA), and they differ considerably in terms of tissue tropism [19,20]. Serovars A, B, Ba, and C are the infectious cause of trachoma which can lead to blindness. Although trachoma is now a thing of the past in the majority of developed countries, it remains a menace in impoverished regions around the globe where there is limited access to healthcare, particularly the Middle East, Asia, and Africa. On the other hand, *C. trachomatis* serovars D, Da, E, F, G, Ga, H, I, Ia, J, and K infect the urogenital tract, whereas L serovars (L1, L2, L2a, and L3) invade the lymphatics and lymph nodes, resulting in lymphogranuloma venereum (LGV). *C. trachomatis* infection has also been linked to reactive arthritis at remote joints that often commences a few weeks following the primary site bacterial infection [21]. Certain members of the *Chlamydia* genus such as *C. pneumoniae* cause respiratory tract infections in humans, whereas other species including *C. felis*, *C. avium*, and *C. abortus*, despite infecting nonhuman animals as their primary host, can cause zoonotic disease in humans [22].

## 2. *Chlamydia* Metabolism

### 2.1. Chlamydia as an Energy Parasite

The host-restricted lifestyle of *C. trachomatis* is reflected in its limited genome of approximately 1.04 Mbp that is deficient in many essential genes necessary for the synthesis of metabolic cofactors, amino acids, lipids, and nucleotides. In virtually all strains of *C. trachomatis* characterized, a ~7.5 kbp plasmid that has been implicated in bacterial virulence is also present [23,24]. The pathogen is unable to carry out de novo biosynthesis of ATP, UTP, and GTP. Nevertheless, *C. trachomatis* has retained a functional CTP synthetase gene which permits the conversion of UTP to CTP [25,26]. Both *Chlamydia* EBs and RBs can survive under axenic conditions with defined nutrients. Whereas EBs generate ATP from the metabolism of glucose-6-phosphate, ATP, and amino acids, RBs scavenge the host ATP. This distinct ability to harness different energy sources ensures the survival of EBs in the extracellular environment prior to infection [27,28,29,30].

For decades, the bacteria in the family Chlamydiaceae were considered as exclusive “energy parasites” that feed on the host ATP to fulfill their demands for energy. This largely stemmed from the early biochemical observations that chlamydial organisms show no demonstrable succinoxidase and cytochrome *c* reductase activities, which pointed to the absence of several critical components of the mitochondrial respiratory chain, i.e., flavoproteins and cytochromes [31,32,33]. Subsequently, it was discovered that *Chlamydia* species possess a pair of nucleotide transporters, Npt1 and Npt2 (nucleoside phosphate transporters 1 and 2), allowing them to siphon nucleotides from their host. This, along with the finding that Chlamydiaceae lack the capability for de novo biosynthesis of nucleotides, with the exception of CTP, further bolstered the energy parasite hypothesis [26,34,35]. Genomic studies in the later years, however, shed a different light on the energy metabolism of *Chlamydia* and uncovered a slew of information on the oftentimes underappreciated metabolic capabilities of these organisms.

### 2.2. Glucose Metabolism in Chlamydia

The genome of *C. trachomatis* underwent substantial reduction in the course of its evolution as it geared toward an intracellular lifestyle, resulting in the loss of many essential genes. Despite this, *C. trachomatis* encodes and expresses a complete pentose phosphate pathway (PPP) and virtually all necessary enzymes to support glycolysis. While this may allow the bacterium to produce ATP and the fundamental reducing agents NADH and NADPH, it lacks a functional hexokinase gene and, therefore, draws on the host’s reserves of glucose in the form of glucose-6-phosphate (G6P) [24,36,37,38,39,40,41,42,43,44,45,46,47]. Consistent with this, experimental results indicate that the EBs and RBs differ in their requirements for energy metabolites, with G6P being the preferred substrate for EBs, whereas ATP is utilized by the RBs [29,48]. ATP is taken up by *Chlamydia* using a dedicated pair of ATP/ADP translocases [26,34,35]. As *Chlamydia* do not possess a phosphotransferase system (PTS) for the uptake of carbohydrates, G6P is instead imported into the bacteria through a sugar-phosphate/inorganic-phosphate antiporter [28,49,50,51].

*C. trachomatis* also possesses a complete gluconeogenic pathway and all components required for glycogen synthesis and glycogenolysis, but it remains unclear to which degree these pathways contribute to their energy metabolism. In fact, *C. trachomatis* appears to lack a functional carbon catabolite repression (CCR) control system; therefore, it is unable to grow well in environments enriched with gluconeogenic precursors such as glutamate, malate, α-ketoglutarate, and oxaloacetate, suggesting that these pathways are unlikely to serve as efficient input of energy [49,50,52,53].

### 2.3. Handicapped TCR Cycle in Chlamydia

A further limitation imposed on chlamydial energy metabolism is its fragmented tricarboxylic acid (TCA) cycle, which is devoid of citrate synthase, aconitase, and isocitrate dehydrogenase, which catalyze the first three reactions of the process. Although there is evidence that *C. trachomatis* can circumvent this shortfall by acquiring part of the TCA intermediates from the host, such as malate (and metabolize it to fumarate and succinate), the proteins succinate dehydrogenase subunit C and fumarate hydratase carry frameshift mutations in their coding genes (*SdhC* and *FumC*), indicating that these TCA enzymes may not operate optimally [54,55,56,57]. This evidence strongly points to a lifestyle of *C. trachomatis* that hinges to a significant extent on its capacity to siphon the energy metabolites from the host.

## 3. Manipulation of Host Mitochondria by *Chlamydia*

### 3.1. Alteration of Mitochondria Dynamics

Recently, several lines of evidence have begun to emerge suggesting that *C. trachomatis* is able to tip the scale of mitochondrial dynamics in favor of fusion by elevating the level of cyclic AMP (cAMP), resulting in AMP-dependent protein kinase (PKA)-dependent phosphorylation of DRP1 and its consequent inhibition. Accordingly, pharmacological inhibition of adenylate cyclase reduces intracellular cAMP, which impairs mitochondrial elongation and attenuates chlamydial growth [58,59]. Analyses of the infection-induced alteration of miRNA expression have incriminated miRNA in the regulation of p53 during chlamydial infection. miR-30c was upregulated in the infected cells, even in the presence of intense oxidative stress, which would otherwise activate p53 and its downstream target DRP1. Overexpression of both p53 and DRP1 led to severe mitochondrial fragmentation, which restricted chlamydial growth [58]. Another study suggested that *Chlamydia* increases mitochondrial fusion by inhibiting the activation of DRP1 and inhibits mitochondrial fission by blocking DRP1 oligomerization [60]. Silencing mitochondria fusion mediator protein or treatment with adenylate cyclase inhibitor (which diminishes mitochondrial elongation) attenuates chlamydia proliferation in cells, suggesting that modification of mitochondrial dynamics maintains a favorable environment for reproduction and growth of *Chlamydia* [59].

### 3.2. Chlamydia Upregulates HKII to Promote Mitochondria Integrity

Further evidence linking metabolic reprogramming to apoptosis regulation during chlamydial infection derives from an investigation into the interplay between hexokinase and mitochondria. In this study, it was shown that chlamydial infection triggers the phosphorylation of one of the downstream nodes of PI3K/AKT signaling cascade, 3-phosphoinositide-dependent kinase 1 (PDPK1), leading to stabilization of MYC and subsequent upregulation of hexokinase II (HKII) [61]. HKII, through its association with the mitochondrial porin voltage-dependent anion channel (VDAC), cooperates with other proteins including glucose transporter (GLUT) as well as adenine nucleotide translocator (ANT) to take part in carbohydrate metabolism by promoting the production of G6P that serves as input substrate for glycolysis, PPP, and TCA cycle [62]. Additionally, HKII-VDAC complex is involved in maintaining the outer mitochondrial membrane (OMM) integrity by interfering with the mitochondrial cell death pathway, in part due to its ability to interact with the antiapoptotic members of the BCL-2 family proteins to inhibit the mitochondrial binding of the pore-forming proapoptotic effectors BAK and BAX [63,64,65]. Consistent with the role of HKII as an important player in the regulation of apoptosis, interrupting the interaction between HKII and the mitochondria sensitizes infected cells to apoptosis [61].

### 3.3. Chlamydia Mediated Interference of Mitochondrial Import Machinery and Protein Composition

For some species of *Chlamydia* such as *C. psittaci*, maintaining a close contact of its inclusion with the mitochondria appears integral to its infectious cycle. Notably, disrupting the interaction of mitochondria with the *C. psittaci* inclusions by inhibiting kinesin, a motor protein that takes part in the movement of mitochondria in certain cell types, restricts the mobilization of mitochondria to close apposition with the inclusions, thus slowing the RB–EB conversion [66,67]. Derre et al. performed a genome-wide RNAi screen and identified two TOM subunits, i.e., Tom40 and Tom20, that are essential for *C. caviae* infection in *Drosophila* cells [68]. Knockdown of Tom40 and Tom20 decreased the infection of *C. caviae* in HeLa-229 cells. Interestingly, infection of Tom40-depleted cells reduced the recruitment of mitochondria to the vicinity of inclusions, which could be a plausible reason for the observed inhibition of *C. caviae* infection in Tom40 knockdown cells [68]. In an overexpression study of *C. trachomatis* genes in a yeast expression system, CT084, a conserved HKD phospholipase D (PLD)-like protein proposed to play a role in chlamydial pathogenesis, drastically impacted the growth of yeast. CT084 has been experimentally demonstrated to have mitochondrial localization in yeast, suggesting that its cytotoxicity is related to its ability to interfere with mitochondrial function [69].

Recent proteomic data have highlighted interactions between five chlamydial inclusion membrane (Inc) proteins (CT005, CT058, CT195, CT556, and CT819) components of the TIM–TOM complex [70]. The interplay between *Chlamydia* and the mitochondrial protein import pathway can fulfill certain functions that likely benefit its survival. For example, the intricate association of *C. psittaci* inclusions between its inclusions and mitochondria may be essential to rewire the mitochondrial-based pathways, possibly to facilitate the acquisition of nutrients and evasion of host cell death. *Chlamydia* has been known to deploy effector proteins into its host cells via the type III secretion system (T3SS) [71]. Conceivably, some of these effectors may be delivered to the mitochondria through the mitochondrial protein import system or become integrated into the mitochondria and influence the function of mitochondria. Such might be the case of *C. caviae* where Tom40 or Tom22 depletion attenuated bacterial infectivity but not production of ATP and apoptosis rate in infected cells, suggesting that alteration of mitochondrial protein import machinery may have an impact on other mitochondria-targeting mechanisms of survival relevance for *C. caviae* [68]. Employing bioinformatics screening and ectopic expression to identify chlamydial proteins with mitochondrial targeting sequence (MTS), Dimond et al. identified five chlamydial proteins (CT132, CT529, CT618, CT642, and CT647) that were localized to the mitochondria. Although the significance of these proteins in the context of bacteria–host interaction was not experimentally explored, three of these five proteins, namely, CT529, CT618, and CT642, which are putative chlamydial Inc proteins, were found to be T3SS-secreted proteins [72]. Analysis of the mitochondrial proteome from cells infected with *C. trachomatis* using liquid chromatography (LC) tandem mass spectrometry (MS) (LC–MS/MS) revealed mitochondrial localization of CT529 and CT618 in infected cells, as well as uncovered infection-induced changes in the mitoproteome composition, including those responsible for mitochondrial dynamics, apoptosis, and metabolism [72]. Additionally, the authors found that the chlamydial protease-like activity factor (CPAF), a secreted serine protease uniquely conserved in chlamydial organisms with a broad substrate specificity [73], and the *Chlamydia* protein associated with death domain (CADD), which is known to induce cell death [74], were significantly enriched in the mitochondrial proteome, suggesting that these proteins could play a role in modulating the mitochondrial processes.

## 4. Chlamydia and Disruption of Host Cell Metabolism

### 4.1. C. trachomatis Steers the Host toward Hypermetabolic State

To search for the key metabolic processes and host proteins that are essential for chlamydial growth and evaluating the suitability of these factors as drug targets in the treatment of *C. trachomatis*, Rother et al. [75] probed the response of HeLa cells toward *C. trachomatis* infection using a combined approach of gas chromatography/mass spectrometry (GC–MS), as well as small interfering RNA (siRNA). Replication of *C. trachomatis* was found to require several glycolytic enzymes including glucose-6-phosphate isomerase (GPI) and 6-phosphofructokinase (PFKM). Furthermore, a concomitant increase in G6P was observed, demonstrating that enhanced host cell glycolysis is essential for *Chlamydia* propagation. *C. trachomatis* also induced significant upregulation of glutamate, pyruvate, and lactate.

Additionally, RNAi screening highlighted the importance of pyruvate dehydrogenase kinase 2 (PDK2) during infection. PDK2 is a mitochondrial enzyme that regulates the activity of the pyruvate dehydrogenase kinase complex (PDC), which is responsible for oxidative decarboxylation of pyruvate to acetyl-CoA in order to enter the TCA cycle during anaerobic metabolism. Phosphorylation by PDK2 inactivates PDC; hence, more intermediates of glycolysis can be redirected to other metabolic pathways such as PPP for de novo nucleotide synthesis, as well as aerobic glycolysis, whereby pyruvate is metabolized to lactate [76]. Increased aerobic glycolysis is most commonly associated with actively proliferating cancer cells [77]. Lactate, the end-product of aerobic glycolysis, can serve as an alternative source of energy for cancer cells. Human cervix squamous carcinoma cells, for instance, are capable of utilizing lactate through oxidation to pyruvate, which can then be channeled back to glycolysis to produce ATP [78]. Thus, in a manner analogous to cancer cells, *C. trachomatis* steers the host to a hypermetabolic state in order to cope with increased metabolic requirements of its proliferation.

### 4.2. Chlamydial Propagation Requires Purine Metabolism

In keeping with the fact that *C. trachomatis* triggers the degradation of p53 to maintain the PPP, several PPP enzymes were identified as crucial host factors for chlamydial growth, namely, glucose-6-phosphate dehydrogenase (G6PD) and thiamine pyrophosphokinase isoform 1 (TPK1) [75]. G6PD is a key enzyme in the oxidative PPP that oxidizes G6P to produce phosphogluconolactone (PG6) [79]. TPK1, on the other hand, catalyzes the phosphorylation of thiamine into thiamine pyrophosphate (TPP), which acts as a cofactor for transketolase (TKT) of PPP [80]. TKT generates ribose-5-phosphate (R5P) in a reversible reaction belonging to the nonoxidative arm of the PPP [81]. In this way, the PPP provides the initial substrate, R5P, required for nucleotide biosynthesis. Indeed, *C. trachomatis* infection elevated R5P and guanosine monophosphate (GMP) in the host, indicating an increase in substrate flux toward the PPP and, subsequently, de novo purine metabolism.

Since the genome of *C. trachomatis* does not encode the complete enzyme paraphernalia involved in nucleotide biosynthesis including GTP, supply of nucleotides must be compensated for by the host [24]. Indeed, four additional host factors that are involved in de novo purine metabolism i.e., phosphoribosyl pyrophosphate synthetase 2 (PRPS2), phosphoribosyl pyrophosphate amidotransferase (PPAT), inosine-5-monophosphate dehydrogenase 2 (IMPDH2), and guanosine monophosphate synthetase (GMPS), were found to be significantly associated with *Chlamydia* propagation. Interference with these enzymes through small hairpin RNA (shRNA) in HeLa and primary ectocervical cells limited the formation of infectious *C. trachomatis* progeny. Notably, mice treated with the pharmacological inhibitor of IMPDH2 i.e., mycophenolate mofetil (MMF) had significantly reduced bacterial load and disease burden upon *C. trachomatis* infection, indicating that successful replication of *C. trachomatis* is tied to a functional host de novo purine metabolism [75].

### 4.3. Low Level of p53 Favors Chlamydial Growth

Infection with *C. trachomatis* is known to activate the PI3K/AKT pathway, which stimulates mouse double min 2 homolog (MDM2)-mediated degradation of the tumor suppressor protein p53 [82,83]. In addition to its well-established role in controlling the onset and progression of cancer development, p53 regulates many different effector genes and, therefore, holds sway over a variety of processes related to apoptosis, cell-cycle progression, DNA repair, cellular senescence, and metabolic adaptation [84]. In the context of metabolism, p53 negatively regulates the rate-limiting enzyme glucose-6-phosphate dehydrogenase (G6PD) of PPP, which generates NADPH required for the production of reduced glutathione (GSH), an antioxidant [85,86]. MDM2 controls the activity of p53, which stably maintains p53 at low levels under normal circumstances. MDM2 inhibits the transcriptional activation of p53. In addition, M2M2 acts as an E3 ubiquitin ligase and promotes the ubiquitination of p53, resulting in p53 nuclear export and condemning p53 to degradation by proteasome [87].

It is known that early-onset *C. trachomatis* infection induces the formation of ROS [88]. If left unchecked, excessive buildup of ROS is detrimental to the cells as it causes damage to DNA in the form of double-stranded breaks and telomere shortening [89,90]; both events have been observed following chlamydial infection [91,92,93]. In the event of severe DNA damage, p53 can trigger the arrest of cell-cycle progress and apoptosis [84]. This poses a threat to *C. trachomatis* as its survival relies on the host cell. Maintaining low levels of p53, therefore, may enable *C. trachomatis* to counteract the cytotoxic stress induced by DNA damage, by shifting the metabolic balance away from glycolysis to PPP to produce NADPH and nucleotides, which are crucial for antioxidant defense and promoting DNA repair. Indeed, downregulation of p53 during chlamydial infection reduced the expressions of both the cell-cycle arrest mediator p21 and the BH3-only proapoptotic protein PUMA. Furthermore, disrupting G6P activity and p53–MDM2 interaction impairs the development of *C. trachomatis*, attesting to the importance of p53 to metabolic reprogramming and apoptosis regulation in infected cells [82,83].

## 5. *Chlamydia* Intervenes in Host Cell Apoptosis

### 5.1. A General Overview of Apoptosis

There are two major pathways of apoptosis, namely, the intrinsic and extrinsic pathways, and a third mode is induced by cytotoxic T lymphocytes (CTLs) and natural killer (NK) cells (Figure 1). These signaling pathways may act in concert or independently to commit the cells to the irreversible fate of death. Irrespective of the route of initiation, the overarching endpoint for all modes of apoptosis is the activation of effector caspases 3, 6, and 7, which execute the proteolytic cleavage of numerous cellular proteins, many of these perform structural and regulatory roles, in order to prepare cells for eventual destruction [94]. The baculovirus IAP repeat (BIR) domain-containing inhibitor of apoptosis proteins (IAPs) family of proteins, notably X-chromosome-linked IAP (XIAP), cellular IAP-1 and -2, (cIAP1 and cIAP2), governs the activity of caspases by directly binding and inhibiting their activation. These proteins also possess a characteristic RING domain that allows them to interact with ubiquitin ligases; thus, they may play a role in the regulation of caspases through the ubiquitin proteasome system [95]. The second mitochondria-derived activator of caspases (SMAC/DIABLO) and serine protease high temperature requirement protein A2 (HtrA2)/OMI, which are both released from the mitochondria during apoptosis, counteract and irreversibly inactivate the IAPs via either the ubiquitin–proteasome pathway or direct proteolysis, thus clearing the way for the apoptosis cascade [96,97].

The extrinsic apoptotic pathway responds to external death stimuli through a number of transmembrane death receptors belonging to the tumor necrosis factor (TNF) superfamily, including Fas/CD95 and TNF-related apoptosis-inducing ligand (TRAIL) receptors. When engaged with their cognate ligands, for instance, TRAIL and FasL/CD95L, the death receptor trimer aggregates which recruits the Fas-associated death domain (FADD) adaptor protein via homotypical interaction of death domains (DD) [98,99]. The Fas-FADD in turn, binds the inactive initiator procaspases -8 and -10. Interactions between the death effector domain (DED) of the FADD and procaspases form the death-inducing signaling complex (DISC) wherein the procaspases dimerize and undergo proximity-induced autocatalysis to yield their active forms [100]. The mature initiator caspases process and activate the effector caspases to carry out various proteolysis events to elicit cell death. In some instances, the extrinsic pathway can convey the apoptotic cues to the intrinsic pathway. The initiator caspases orchestrate this process by cleaving the BH3-interacting domain death agonist (BID) to generate truncated BID (tBID), which then makes its way to the mitochondria to amplify apoptosis via the intrinsic pathway [101,102].

A variety of cytotoxic cellular events including organelle damage, expression of oncogenes, deprivation of cytokines and growth factors, developmental cues, and chemotherapy drugs can initiate the intrinsic pathway. When sufficiently activated, the BH3-only proteins bind and induce conformational changes in BAK and BAX to result in the formation of oligomeric pores on the OMM, leading to mitochondrial outer membrane permeabilization (MOMP). This loss of OMM integrity causes the release of apoptogenic factors from the intermembrane space into the cytosol, including cytochrome *c*, SMAC/DIABLO, and HtrA2/OMI [103]. Cytochrome *c* mediates the heptameric assembly of the apoptosome by binding to the apoptotic protease-activating factor 1 (APAF1) and dATP. Recruitment of monomeric procaspase 9 by the apoptosome and subsequent dimerization-induced activation produce the mature caspase 9 necessary to kick start the effector caspase cascade [104]. The intrinsic pathway is tightly regulated by diverse members of the BCL-2 family of proteins that are defined by the presence of one or multiple BCL-2 homology (BH) domains (BH1–BH4). These proteins are categorized into three distinct groups, i.e., prosurvival members (BCL-2, BCL-B, BCL-X_L_, BCL-W/BCL-2-L2, BFL1/BCL2-A1, and MCL-1), the BH3-only proapoptotic initiator proteins (BAD, BID, BIK, BIM, BMF, HK, NOXA, and PUMA), and the pore-forming proapoptotic effector proteins (BAK, BAX, and BOK) [103,105,106].

CTLs and NK cells provoke apoptosis via cytolytic molecules, i.e., perforin and granzyme B. Once released from the secretory granules of CTL or NK cell, perforin permeabilizes the target cell membrane by forming large pores to allow entry of granzyme B into the cell [107]. As with caspases, granzyme B is a serine protease that enables it to communicate with the cell death pathway in two distinct ways. It is able to directly activate the initiator caspases (8 and 10), and or the downstream effector caspases (3, 6, and 7). Additionally, it can employ an indirect approach to instigate the caspase chain reaction by cleaving the mitochondrial outer membrane permeabilization mediator BID to bring about cell killing through the intrinsic apoptosis route [108,109,110].

**Figure 1 microorganisms-11-01382-f001:**
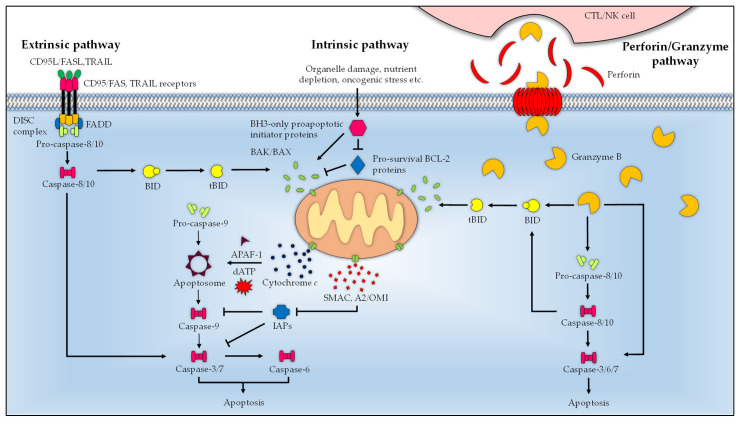
Schematic summary of the major apoptosis pathways. Engagement of ligand (Fas/CD95L and TRAIL) with the transmembrane death receptors (Fas/CD95 and TRAIL receptors) activates the extrinsic pathway and recruits FADD. The Fas-FADD binds the procaspases 8 and 10, which leads to the formation of DISC where the procaspases mature. The active initiator caspases 8 and 10 proteolytically activate the effector caspases (3, 6, and 7) to execute apoptosis. In the intrinsic pathway, stimuli-activated BH3-only proapoptotic initiator proteins bind the pore-forming proapoptotic effector proteins BAK and BAX to induce MOMP, which results in the cytosolic release of cytochrome *c*, SMAC/DIABLO, and HtrA2/OMI [101]. Cytochrome *c,* in the presence of dATP, mediates the assembly of the monomeric APAF1 into the heptameric apoptosome complex, which processes and activates procaspase 9 into the mature caspase 9 to drive the downstream apoptosis cascade. The activity of the proapoptotic effector proteins is regulated by the pro-survival BCL-2 proteins (BCL-2, BCL-B, BCL-XL, BCL-W/BCL-2-L2, BFL1/BCL2-A1, and MCL-1), which are in turn controlled by BH3-only proapoptotic initiator proteins (BAD, BID, BIK, BIM, BMF, HK, NOXA, and PUMA). BID bridges the intrinsic and extrinsic pathway, which, upon proteolytic processing by the initiator caspases (8 and 10) into tBID, translocates to the mitochondria where it activates the intrinsic pathway. The perforin/granzyme pathway is initiated by perforin and granzyme B of CTL and NK cells. Upon delivery to the target cell, perforin oligomerizes, forming a transmembrane pore on target cells to allow cellular entry of granzyme B. Granzyme B acts as a serine protease by activating the initiator (8 and 10), and or effector (3, 6, and 7) caspases. Alternatively, granzyme B cleaves BID to tBID, and liaises with the intrinsic pathway to trigger apoptosis.

### 5.2. Inactivation of BAK and BAX Apoptotic Molecules

Research examining *Chlamydia*-infected cells has consistently shown that chlamydial infection confers resistance to various intrinsic and extrinsic proapoptotic stimuli, as well the granzyme B/perforin pathway, which exhibited characteristic inhibition of cytochrome *c* release [111,112,113,114,115]. The importance of mitochondria in the modulation of host cell apoptosis by *Chlamydia* is highlighted in experiments with Fas-expressing type I and type II cells. Rather than type I cells where apoptosis proceeds independently of mitochondria, it was found that *Chlamydia* species were able to protect type II cells from apoptosis, requiring a mitochondrial step to activate the effector caspases. This indicates that *Chlamydia* mediates the antiapoptotic actions through mechanisms that are directed toward the mitochondrial signaling pathways [116].

One likely strategy that *Chlamydia* uses to preserve the mitochondrial function is by blocking the activation of the proapoptotic BH3-only effector proteins BAK and BAX. In addition, *Chlamydia* appears to degrade BH3-only proapoptotic initiator proteins BAD, BIK, BIM, and PUMA, likely accounting for the lack of BAK and BAX activation and subsequent cytochrome *c* release during chlamydial infection [117,118,119,120]. Early reports identified CPAF as the chlamydial effector responsible for the cleavage of a number of these BH3-only proteins, including BIK, BIM, and PUMA. However, subsequent investigations of an extensive list of reported CPAF substrates encompassing BIK, BIM, and PUMA showed that these were likely artefacts of the experimental procedures due to post-cell-lysis degradation [121,122]. A subsequent study utilizing CPAF-deficient mutant cells lent further support to this observation, which did not produce a significant impact on cell viability [123]. However, it remains a question at this point whether CPAF represents a case of redundancy that plays a role in supporting the overall anti-apoptosis phenotype of infected cells. A recent study by Kontchou et al. provided interesting insight into the basis of chlamydial apoptosis inhibition and showed that *C. trachomatis* disrupts the interaction of BAK/BAX with VDAC2 to prevent their activation [124]. In this scenario, *C. trachomatis* blocks the release of BAK from VDAC2; as VDAC2 is known to regulate the activity of BAK by binding to BAK to restrict apoptosis [125], this inhibits the oligomerization of BAK, thus hindering the activation of apoptosis. Intriguingly, the chlamydial OmpA was found to associate with the mitochondria when overexpressed in uninfected human cells, and it protected cells from apoptosis stimulation, an effect largely mirroring *C. trachomatis*-infected cells, suggesting OmpA as a chlamydial antiapoptotic effector [124].

### 5.3. Inhibition of IAP Molecules

*Chlamydia* can target the IAPs to inhibit apoptosis via the upregulation of cIAP-2. This process requires the presence of other members of IAPs, namely, cIAP-1, XIAP, and survivin. Although the precise mechanism is not known, it is believed that interaction of cIAP-1, cIAP-2, XIAP, and survivin in infected cells forms a heteromeric complex that impedes the effector caspases, thereby attenuating apoptosis [126,127,128,129]. Enhanced expression of cIAP-2 seen during *C. pneumoniae* infection requires functional NF-κB signaling [127,128,130]. In comparison, *C. trachomatis*-infected cells challenged with proapoptotic stimuli did not lead to detectable NF-κB activity. Additionally, pharmacological blockade and deletion of NF-κB did not abolish the infection-induced apoptotic resistance, suggesting that NF-κB is likely dispensable for the antiapoptotic activity of *C. trachomatis* [129]. Nonetheless, the exact relevance of this pathway to other *Chlamydia* in the context of apoptosis modulation is not known.

### 5.4. Interference with BAD and PI3K/AKT Pathway

*Chlamydia* infection-associated activation of PI3K has been shown to depend on the host cell surface tyrosine kinase EphrinA2 receptor (EphA2). EphA2 attaches and mediates *Chlamydia* entry into host cell, wherein it remains bound to internalized bacteria and activates PI3K. ERK activation during the mid-phase of *C. trachomatis* infection stimulates the expression of EphA2, which is trafficked to and associated with inclusion. This inclusion-bound EphA2 activates PI3K, which in turn activates AKT. Noteworthily, EphA2 knockdown promoted cell apoptosis, indicating that EphA2-mediated signaling plays a supportive role in *Chlamydia* infection-induced apoptosis resistance [131]. *C. trachomatis* intervenes in the mitochondrial translocation of BAD in a PI3K/AKT-dependent manner. This involves activation of the PI3K/AKT pathway and subsequent phosphorylation of BAD. The phosphorylated BAD is then recruited to the chlamydial inclusion surface where it binds to a cellular protein 14-3-3β that associates with the chlamydial inclusion membrane protein IncG, thereby inhibiting the interaction between BAD and mitochondria [132]. Activated PKCδ can translocate to the mitochondria to trigger apoptosis [133,134]. *C. trachomatis* can sequester PKCδ similarly by diverting the enzyme from mitochondria to the vicinity of the inclusion that is enriched in diacylglycerol (DAG), possibly through a mechanism that requires the C1 domain of PKCδ [135]. In a similar fashion, *Chlamydia* can sequester caspase 9 into the inclusion which confers a block at the downstream caspase cascade, thus putting the brakes on apoptosis [136].

### 5.5. Stabilization of the Apoptosis Regulator MCL-1

Elevation of the pro-survival BCL-2 family member MCL-1 early during *C. trachomatis* infection has been found to be mediated via the Raf/MEK/ERK (MAPK/ERK) pathway and stabilized through the PI3K/AKT pathway [137,138]. In neutrophils infected with *C. pneumoniae*, activation of the Raf/MEK/ERK and PI3K/AKT pathways accompanies NF-κB-dependent release of IL-8, which further augments MCL-1 expression [138]. Accordingly, increased expression of MCL-1 blocks the release of SMAC, which would otherwise antagonize the activity of IAPs and subsequent activation of effector caspases [137]. Induction of the hypoxia-inducible factor-1α (HIF-1α) plays a role in the maintenance of MCL-1 during the initial stages of chlamydial infection, and likely involves the ERK pathway, as pharmacologically inactivating ERK signaling in infected cells with the MEK-1 inhibitor UO126 could prevent HIF-1α from degradation. At the later phase of infection, the level of HIF-1α decreases sharply, suggesting that *Chlamydia* employs other mechanisms to preserve the expression of MCL-1 during infection [139,140]. Indeed, a study by Fischer and colleagues showed that the chlamydial effector protein ChlaDub1 deubiquitinates MCL-1, which protects MCL-1 from proteasomal degradation. However, results obtained from cells infected with *C. trachomatis ChlaDub1*-null mutant did not exhibit a significant decrease in the MCL-1 protein levels. It is, therefore, possible that other redundant mechanisms to maintain the pool of MCL-1 might be at play [141]. In Hela cells persistently infected with *C. psittaci*, ERK1/2 pathway-dependent upregulation of MCL-1 expression similarly conferred an apoptosis resistance phenotype [142]. Kontchou et al. recently showed that infected cells deficient in MCL-1, BCL-W, and BCL-X_L_, as well as cells pretreated with inhibitors targeting these pro-survival proteins, remained resistant to apoptosis induction, suggesting that these proteins are not essential to chlamydial evasion of host cell apoptosis, and that the loss of MCL-1 could be compensated for by other antiapoptotic mechanisms operating in tandem during chlamydial infection [124].

### 5.6. Upregulation of BAG-1

Infection-mediated activation of MAPK/ERK kinase could also induce BCL-2 associated anthanogene 1 (BAG-1) expression [143]. BAG-1 is a known antiapoptotic protein capable of binding to BCL-2 to delay host cell death and activating the serine/threonine kinase Raf-1, which operates upstream of the MAPK/ERK pathway [144,145]. Inhibition of Raf-1 reduced the level of BAG-1 during chlamydial infection. Depletion of BAG-1 was correlated with the enhanced apoptosis rate of *C. trachomatis*-infected cells. On this basis, it has been suggested that the overall apoptosis resistance during chlamydia infection partly relies on this regulator of apoptosis [143].

## 6. The Influence of *Chlamydia* Plasmid Pgp3 on the Fate of Host Cells

### 6.1. Modulation of the Host Cell Autophagy and Apoptosis

Despite differing tissue tropism, most members of the family Chlamydiaceae, namely, *C. trachomatis, C. muridarum, C. pneumoniae, C. psittaci, C. suis, C. caviae, C. felis, C. gallinacea,* and *C. pecorum*, naturally bear a ~7.5 kb plasmid that is varyingly conserved (69–99%) amongst the species [146,147,148,149]. Although several isolates of *C. trachomatis* devoid of the plasmid have been characterized [150,151,152], the rarity of these plasmid-free variants suggests the presence of a strong selective pressure favoring the persistence of plasmid in the environment. Interestingly, Pgp3 is the only secreted chlamydial plasmid protein that has a cytosolic localization during infection, suggesting its possible role in the manipulation of host cell processes [153]. Pgp3 induces secretion of proinflammatory cytokines including interleukin-6 and interleukin-8 in epithelial cells, and plasmid presence has been associated with a severe pathological outcome in patients [13,154].

While much remains unknown concerning the functional significance of Pgp3 in the biology of *Chlamydia*, emerging evidence has incriminated Pgp3 as a chlamydial effector involved in the regulation of host apoptotic pathway. In a recent study, Lei et al. found that the protein high-mobility group box 1 (HMGB1) was upregulated in serum-starved HeLa cells following Pgp3 stimulation [155]. Moreover, Pgp3 triggered the expression of the BH3-containing protein Beclin-1 and the antiapoptotic BCL-2, which correlated with increased autophagic activity, specifically mitophagy, and reduced apoptotic rate in stimulated cells [155]. HMGB1 competitively displaces the binding of Beclin-1 from BCL-2 by mediating the phosphorylation of BCL-2 and binds to Beclin-1 to promote its autophagic function during nutrient starvation. Reduction of HMGB1 is known to trigger apoptosis owing to sustained Beclin-1–BCL-2 interaction [156]. Indeed, artificially inhibiting autophagy in HeLa cells stably transfected with Pgp3 decreased the expression of BCL-2 and resensitized cells to apoptosis [157].

### 6.2. Interaction with MDM2-p53

The role of Pgp3 became more evident when it was shown to be involved in the phosphorylation of MDM2 at serine-166 that is dependent on the PI3K pathway [158], which is essential for nuclear entry of MDM2 to interact with p53 [159]. Activation of the PI3K/AKT-mediated MDM2–p53 axis induced p53 degradation and attenuated cellular apoptosis, which could be reduced significantly by pharmacological inhibition of the MDM2–p53 interaction [158]. *Chlamydia* infection in HeLa epithelial cells is known to induce profound changes in the host proteome [160]. On a quest to search for the host proteins that interact with Pgp3, Li, Chen, Zhong, Wang and Zhong [153] employed isobaric tags in a high-throughput quantitative proteomics approach on Pgp3-transfected HeLa cells and uncovered a slew of proteins that were upregulated following Pgp3 expression. Among these, HMGB1 and the protein deglycase DJ-1 (otherwise known as Parkinson disease protein 7; PARK7) were significantly expressed [161]. DJ-1 exerts a plethora of roles that include cell survival and proliferation. These functions are attributed to its ability to modulate diverse cellular pathways such as PI3K/AKR and ERK1/2 [162]. DJ-1 can protect cells against the TRAIL-induced extrinsic apoptotic pathway by preventing the formation of DISC [163]. Pgp3 undermines the host apoptosis by increasing the expression of DJ-1 through the ERK1/2 signaling pathway. Accordingly, ablation of DJ-1 led to a reduction in ERK1/2 phosphorylation and predisposed Pgp3-transfected HeLa cells to apoptosis [164]. A mechanistic summary depicting the various strategies undertaken by *C. trachomatis* to subvert the host apoptotic machinery is put forward in Figure 2.

## 7. Conclusions

As one of the most common bacterial STIs with increasing incidence in recent years, genital *C. trachomatis* infection represents a severe threat to public health afflicting the female population of childbearing age. As an obligate intracellular pathogen, *C. trachomatis* relies very much on its host cells to provide nutrients and other building blocks for propagation and survival. To this end, *C. trachomatis* deploys a well-rounded arsenal of strategies to maneuver the host metabolism and alter the cell fate. While chlamydial infections in animals and humans can be readily treated with antibiotics, identification of the tetracycline resistance gene in the swine pathogen *C. suis* has raised considerable concern about the potential spread of antibiotic resistance among the chlamydial species due to its zoonotic potential [165]. A more prudent way forward, therefore, would be the development of alternative and novel therapeutic regimens. Eventually, a safe and effective vaccine may be required to provide a long-term solution to these persistent pathogens. Understanding the host–pathogen interaction is, hence, essential to achieve this goal.

## Figures and Tables

**Figure 2 microorganisms-11-01382-f002:**
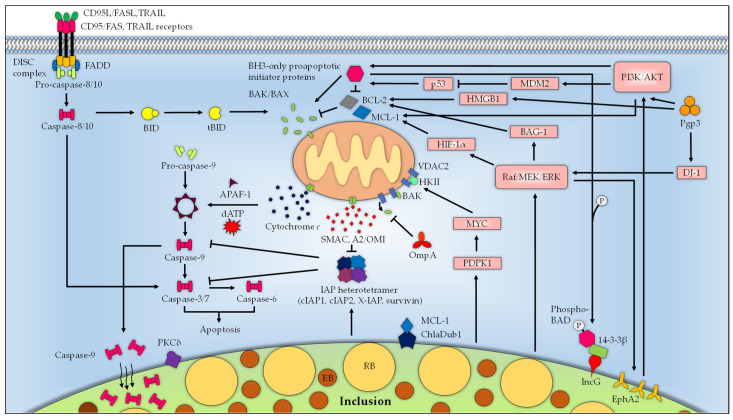
The mitochondrion is the centerpiece in *Chlamydia*-mediated apoptosis resistance. Infection with *Chlamydia* activates the PI3K/AKT and Raf/MEK/ERK pathways to promote host cell survival. Co-internalization of *Chlamydia* EB with the host surface receptor EphA2 into the host cell triggers the activation of PI3K. Raf/MEK/ERK activation during *Chlamydia* infection enhances EphA2 expression, which activates AKT and sustains PI3K/AKT activation. Infection-mediated activation of PI3K/AKT leads to phosphorylation of the BH3-only protein BAD, which is diverted from the mitochondria to the inclusion protein IncG by 14-3-3β. Raf/MEK/ERK also upregulates the expression of BAG-1, which may interact with and enhance the antiapoptotic effect of BCL-2. PI3K/AKT signaling Raf/MEK/ERK stabilizes the pro-survival BCL-2 protein MCL-1. Raf/MEK/ERK-induced expression of HIF-1α maintains the MCL-1 expression. Activated PDPK1 during *Chlamydia* infection signals through MYC to promote the interaction between HKII and VDAC, thereby preventing BAK/BAX-induced apoptosis. Upregulation of IAPs and formation of a heterotetrameric IAP complex consisting of cIAP1, cIAP2, X-IAP, and survivin during chlamydial infection hinders the activation of effector caspases. *Chlamydia* infection additionally sequesters Caspase-9 within the inclusion and redirects PKCδ from the mitochondria to the inclusion to prevent apoptosis induction. Several chlamydial effector proteins contribute to the apoptosis resistance phenotype. ChlaDub1 deubiquitinates MCL-1 and protects it from proteasomal degradation, while OmpA interferes with the release of BAK from VDAC2 to inhibit BAK activation. The *Chlamydia* plasmid-encoded protein Pgp3, on the other hand, upregulates the Raf/MEK/ERK survival pathway through DJ-1 to promote host cell survival. In addition, Pgp3 impedes p53-mediated apoptosis and stimulates MDM2-mediated degradation of p53 by activating the PI3K/AKT pathway. Furthermore, Pgp3 upregulates the autophagic sensor HMGB1 and the antiapoptotic BCL-2 to restrict apoptosis, possibly due to autophagy-induced suppression of apoptosis.

## Data Availability

Not applicable.

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
