# Peer review of "Chlamydia Infection Remodels Host Cell Mitochondria to Alter Energy Metabolism and Subvert Apoptosis"

_microorganisms, 2023, doi:10.3390/microorganisms11061382_

Round 1

Reviewer 1 Report

The review by Cheong and colleagues inadeptly discusses the various strategies utilized by   Chlamydia in manipulating cell metabolism to benefit the bacterial propagation and survival through close interaction with the host cell mitochondrial and apoptotic pathway molecules.

It might be helpful to add a few figures showing the different pathways. 

Author Response

Point 1: The review by Cheong and colleagues inadeptly discusses the various strategies utilized by Chlamydia in manipulating cell metabolism to benefit the bacterial propagation and survival through close interaction with the host cell mitochondrial and apoptotic pathway molecules.

It might be helpful to add a few figures showing the different pathways.

Response 1: We thank you for taking the time to review the manuscript and the feedback.  We have added wo figures as suggested. The first one describes the apoptosis processes on Page 8, with the figure legend that reads as follow (Line 361-380):

“Figure 1: Schematic summary of the major apoptosis pathways. Engagement of ligand (Fas/CD95L and TRAIL) with the transmembrane death receptors (Fas/CD95 and TRAIL receptors) activates the extrinsic pathway and recruits FADD. The Fas-FADD binds the pro-caspases -8 and -10, which leads to the formation of DISC where the pro-caspases maturate. The active initiator caspases -8 and -10 then catalyze the activation of effector caspases (-3,-6,-7) to execute apoptosis. In the intrinsic pathway, activated BH3-only proapoptotic initiator proteins bind the pore-forming proapoptotic effector proteins BAK and BAX to induce MOMP, which results in the cytosolic release of cytochrome c, SMAC/DIABLO, as well as HtrA2/OMI [101]. Cytochrome c, in the presence of dATP, mediates the assembly of the monomeric APAF1 into the heptameric apoptosome complex, which processes and activates pro-caspase 9 into the mature caspase-9 to drive the downstream apoptosis cascade. The activity of the proapoptotic effector proteins are regulated by the prosurvival BCL-2 proteins (BCL-2, BCL-B, BCL-XL, BCL-W/BCL-2-L2, BFL1/BCL2-A1, and MCL1), which are in turn controlled by BH3-only proapoptotic initiator proteins (BAD, BID, BIK, BIM, BMF, HK, NOXA, and PUMA). BID bridges the intrinsic and extrinsic pathway, which upon proteolytic processing by the initiator caspases (-8 and -10) into tBID, translocates to the mitochondria where it activates the intrinsic pathway. The perforin/granzyme pathway is initiated by perforin and granzyme B of CTL and NK cell. Upon delivery to the target cell, perforin oligomerizes, forming transmembrane pore on target cell to allow cellular entry of granzyme B. Granzyme B acts as a serine protease by activating the initiator (-8 and -10), and or effector caspases (-3, -6, and -7). Alternatively, granzyme B cleaves BID to tBID, and liaises with the intrinsic pathway to trigger apoptosis.”

The second figure on Page 12 illustrates the various mechanisms employed by Chlamydia to modulate the host cell apoptosis, with the following figure legend (Line 531-552):

Figure 2. The mitochondrion is the centerpiece in Chlamydia-mediated apoptosis resistance. Infection with Chlamydia activates the PI3K/AKT and Raf/MEK/ERK pathways to promote host cell survival. Co-internalization of Chlamydia EB with the host surface receptor EphA2 into host cell triggers the activation of PI3K. Raf/MEK/ERK activation during Chlamydia infection enhances EphA2 expression, which activates AKT and sustains PI3K/AKT activation. Infection-mediated activation of PI3K/AKT leads to phosphorylation of the BH3-only protein BAD, which is diverted from the mitochondria to the inclusion protein IncG by 14-3-3β. Raf/MEK/ERK also upregulates the expression of BAG-1, which may interact with and enhance the anti-apoptotic effect of BCL-2. PI3K/AKT signaling Raf/MEK/ERK stabilizes the pro-survival BCL-2 protein MCL-1. Raf/MEK/ERK-induced expression of HIF-1α maintains the MCL-1 expression. Activated PDPK1 during Chlamydia infection signals through MYC to promote the interaction between HKII and VDAC, thereby preventing BAK/BAX induced apoptosis. Upregulation of IAPs and formation of heterotetrameric IAP complex consisting of cIAP1, cIAP2, X-IAP, and survivin during chlamydial infection hinders the activation of effector caspases. Chlamydia infection additionally sequesters Caspase-9 within the inclusion and redirects PKCδ from the mitochondria to the inclusion to prevent apoptosis induction. Several chlamydial effector proteins contribute to the apoptosis-resistance phenotype. ChlaDub1 deubiquitinates MCL-1 and protects it from proteasomal degradation while OmpA interferes with the release of BAK from VDAC2 to inhibit BAK activation. The Chlamydia plasmid-encoded protein Pgp3, on the other hand, upregulates the Raf/MEK/ERK survival pathway through DJ-1 to promote host cell survival. In addition, Pgp3 impedes p53-mediated apoptosis and stimulates MDM2-mediated degradation of p53 by activating the PI3K/AKT pathway. Furthermore, Pgp3 upregulates the autophagic sensor HMGB1 and the anti-apoptotic BCL-2 to restrict apoptosis, possibly due to autophagy-induced suppression of apoptosis.”

Reviewer 2 Report

This review by Cheong et al. gives a detailed and in-depth overview over an important part of chlamydia-host cell interactions. It is a very interesting compilation of informations.

There are only one issue concerning the objective stated by the authors:

The authors focus on the energy metabolism and apoptosis. The authors state that this knowledge is important for the development of vaccines. However, the development of vaccines is rather based on immunological responses by the host and should prevent infection. Interestingly they mention repeatedly that blockage of certain pathways is able to reduce proliferation of chlamydia. Thus, this knowledge may be rather important for treatment procedures. Thus, the goal of this review should be reconsidered and restated.

1.    The text is sometimes difficult to understand, because some sentences are exceedingly long convoluted.

2.    The manuscript needs revision of the English language. In the following some problems are indicated:

L16: … cells …

L25:  … stretching …

L34: … as scrapings obtained …

L35: delete similar

L36: … considered to be …

L38: … debate about the pathogen´s identity …

L39: …entity to the ….

L47: … females …

L48: … An increasing …

L51: …. infections ….

L53/54: … of most infections ….

L55: … spread to …

L70: … at remote joints that often commences ….

L73: … abortus which use animals as …

L73: this list should include C. psittaci and C. gallinacea

L74: … host, but have zoonotic potential

L79: … metabolic cofactors ….

L87: … the survival of EBs …

L95: … transporters, Npt1 ….

L117: … required for … and …. unclear to which degree these pathways ….

L120: … is unable …

L121: … … like glutamate, 

L127: … by acquiring part …

L139: “In the absence of increase …” This is unclear; please rephrase.

L146: … suggesting that modification …

L147: … mitochondrial …

L155, L302, L309, L315: through!

L181: … suggesting that its …

L187/188: … with mitochondria ….

L190: … its host cell via …

L 193: … mitochondria and …

L196: … mitochondria-targeting …

L199: … hypermetabolic …

L201: … as drug targets …

L205: … including glucose- ….

L207: … host cell glycolysis …

L210: … screening …

L214: delete: normal

L265: … leads …

L267: … abortus …

L273: … cycle progress …

L274: … a threat … on the host cell.

L283: … intervenes with host …

L288: “No matter the route ……” This is unclear; please rephrase.

L295: … that allow them …

L347: … independent …

L348: “Manipulating the ….” This is unclear; please rephrase.

L355: …. chronic …

L359: … of host cell ….

L378: … mutant cells lent ….

L387: … forms …

L403: … BAD …

L407: … inclusions, which confers …

L416: … of a strong … and … favoring the persistence of plasmid.

L443: … high-throughput …

Author Response

Point 1: This review by Cheong et al. gives a detailed and in-depth overview over an important part of chlamydia-host cell interactions. It is a very interesting compilation of informations.

 Response 1: We thank you for the positive appraisal. 

There are only one issue concerning the objective stated by the authors:

Point 2: The authors focus on energy metabolism and apoptosis. The authors state that this knowledge is important for the development of vaccines. However, the development of vaccines is rather based on immunological responses by the host and should prevent infection. Interestingly they mention repeatedly that blockage of certain pathways is able to reduce proliferation of chlamydia. Thus, this knowledge may be rather important for treatment procedures. Thus, the goal of this review should be reconsidered and restated.

Response 2: We thank you for the suggestions. We have revised the conclusion as follows (Page 12, Line 553-566).

“As one of the most common bacterial STI with increasing incidence in recent years, genital C. trachomatis infection represents a severe threat to public health afflicting the female population of the childbearing age. As an obligate intracellular pathogen, C. trachomatis relies very much on its host cells to provide nutrients and other building blocks for propagation and survival. To this end, C. trachomatis deploys a well-rounded arsenal of strategies to maneuver the host metabolism and alter the cell fate. While chlamydial infections in animals and humans can be readily treated with antibiotics, the identification of tetracycline-resistance gene in the swine pathogen C. suis has raised considerable concern about the potential spread of antibiotic resistance among the chlamydial species due to its zoonotic potential [166]. A more prudent way forward, therefore, would be the development of alternative and novel therapeutic regimens. Eventually, a safe and effective vac-cine may be required to provide a long-term solution to these persistent pathogens. Understanding host-pathogen interaction is hence essential to achieve this goal.”

Point 3: Comments on the Quality of English Language

The text is sometimes difficult to understand, because some sentences are exceedingly long and convoluted.

The manuscript needs revision of the English language. In the following some problems are indicated:

L16: … cells …

L25:  … stretching …

L34: … as scrapings obtained …

L35: delete similar

L36: … considered to be …

L38: … debate about the pathogen´s identity …

L39: …entity to the ….

L47: … females …

L48: … An increasing …

L51: …. infections ….

L53/54: … of most infections ….

L55: … spread to …

L70: … at remote joints that often commences ….

L73: … abortus which use animals as …

L73: this list should include C. psittaci and C. gallinacea

L74: … host, but have zoonotic potential

L79: … metabolic cofactors ….

L87: … the survival of EBs …

L95: … transporters, Npt1 ….

L117: … required for … and …. unclear to which degree these pathways ….

L120: … is unable …

L121: … … like glutamate,  …

L127: … by acquiring part …

L139: “In the absence of increase …” This is unclear; please rephrase.

L146: … suggesting that modification …

L147: … mitochondrial …

L155, L302, L309, L315: through!

L181: … suggesting that its …

L187/188: … with mitochondria ….

L190: … its host cell via …

L 193: … mitochondria and …

L196: … mitochondria-targeting …

L199: … hypermetabolic …

L201: … as drug targets …

L205: … including glucose- ….

L207: … host cell glycolysis …

L210: … screening …

L214: delete: normal

L265: … leads …

L267: … abortus …

L273: … cycle progress …

L274: … a threat … on the host cell.

L283: … intervenes with host …

L288: “No matter the route ……” This is unclear; please rephrase.

L295: … that allow them …

L347: … independent …

L348: “Manipulating the ….” This is unclear; please rephrase.

L355: …. chronic …

L359: … of host cell ….

L378: … mutant cells lent ….

L387: … forms …

L403: … BAD …

L407: … inclusions, which confers …

L416: … of a strong … and … favoring the persistence of plasmid.

L443: … high-throughput …

Response 3: We thank you for pointing out these mistakes. We have corrected all mistakes as indicated. Additionally, we have reviewed the English used in the manuscript and changed the sentences to more appropriate lengths to avoid causing confusion to the readers. Once again, we thank you for your suggestions.

Reviewer 3 Report

This is quite an extensive review of chlamydial interactions with mitochondria.  I really only have one suggestion.  The review overlooks a fairly recent manuscript that speaks directly to the title of the review.  ‘Chlamydia trachomatis Alters Mitochondrial Protein Composition and Secretes Effector Proteins That Target Mitochondria’ (PMID: 36286535) was published in Dec. 2022 and is definitely relevant to the review.  Otherwise, the review contains a lot of good information that could be useful to the field. 

Author Response

Point 1: This is quite an extensive review of chlamydial interactions with mitochondria.  I really only have one suggestion.  The review overlooks a fairly recent manuscript that speaks directly to the title of the review.  ‘Chlamydia trachomatis Alters Mitochondrial Protein Composition and Secretes Effector Proteins That Target Mitochondria’ (PMID: 36286535) was published in Dec. 2022 and is definitely relevant to the review.  Otherwise, the review contains a lot of good information that could be useful to the field.

Response 1: We thank you for the positive comment. We took note on the suggestion and have added the information on the indicated manuscript on Page 5, Line 201-217, and reads as follows:

“Employing bioinformatics screening and ectopic expression to identify chlamydial proteins with mitochondrial targeting sequence (MTS), Dimond et al., identified five chlamydial proteins (CT132, CT529, CT618, CT642, as well as CT647) that were localized to the mitochondria. Although the significance of these proteins in the context of bacteria-host interaction was not experimentally explored, three of these five proteins, namely CT529, CT618, and CT642, which are putative chlamydial Inc proteins, were found to be T3SS secreted proteins. Analysis of mitochondrial proteome from cells infected with C. trachomatis using liquid chromatography (LC) tandem mass spectrometry (MS) (LC-MS/MS) revealed mitochondrial localization of CT529 as well as CT618 in infected cells, and uncovered infection-induced changes in the mitoproteome composition, including those responsible for mitochondrial dynamics, apoptosis, and metabolism. Additionally, the authors found that the chlamydial protease-like activity factor (CPAF); a secreted serine protease uniquely conserved in chlamydial organisms with a broad substrate specificity [71], and the Chlamydia protein associated with death domain (CADD), which is known to induce cell death [72], were significantly enriched in the mitochondrial proteome, suggesting that these proteins could play a role in modulating the mitochondrial processes."